



# Phylogeny of the Stipa and implications for grassland evolution in China: based on biogeographic evidence

**Qing Zhang[1*], Junjun Chen[1], Yong Ding[2]**
[1] School of Ecology and Environment, Inner Mongolia University, Hohhot, 010021,
China
[2] Grassland Research Institute of Chinese Academy of Agricultural Sciences, Hohhot,
010010, China;
* Corresponding author. Email: qzhang82@163.com
**Abstract**
The evolution of Chinese grassland is a still an important question biogeography. In this study, the
phylogeny of 20 Stipa species (extensively distributed in Chinese grassland) was established to
explore the origin and dispersal routes of Chinese grassland. It showed that Stipa species
originated at 28 MaBP and they started to differentiate in central Inner Mongolia at 22 MaBP.
Then, Stipa species expanded along four routes: (1) they expanded from central Mongolia to the
Qilian Mountains, Qinghai, and western Tibet at 16 MaBP. They then gradually expanded from
western to eastern Tibet from 11-6 MaBP. (2) At 12 MaBP, they expanded from central Inner
Mongolia to the Helan Mountains. (3) At 8 MaBP, they expanded from central Inner Mongolia to
the Xinjiang area. (4) At 4 MaBP, they expanded from central to eastern Inner Mongolia.
Therefore, we could deduce the formation order of Chinese grasslands: central Inner Mongolia >
Qilian Mountains > Qinghai > western Tibet > Helan Mountains > Xinjiang > central Tibet >
eastern Tibet > eastern Inner Mongolia. We highlight the importance of the uplift of the
Qinghai-Tibet Plateau and paleoclimate changes in triggering the origin and evolution of Stipa
species and Chinese grasslands.
**Keywords:** uplift of the Qinghai-Tibet Plateau; paleoclimate; explosive radiation;
phylogeny




**1 Introduction**

Different regions on earth harbor distinct biological species even though their
environment are the same, indicating that the distribution of organisms not only is
driven by contemporary ecological factors but may also be closely associated with
historical factors (e.g., geological evolutionary history and population evolutionary
history) (Medail and Diadema, 2009; Ordonez and Svenning, 2015; Santos et al.,
2017). Based on molecular clocks, the genetic information of existing species could
be used to explore the biogeographic processes including the origin, divergence,
expansion, and isolation of organisms at the molecular level during all important
geological history events. In addition, the time of divergence between closely related
species could be basically confirmed (Tedesco et al., 2017; Voskamp et al., 2017;
Wang et al., 2017a).
Grasslands are an important component of the global ecosystem and provide
significant ecosystem functioning and service.. One sixth total land area is covered by
grassland (Scurlock and Hall, 1998). The Chinese grasslands, starting from the
Northeast China Plain, Inner Mongolian Plateau, Ordos Plateau, and Loess Plateau to
the southern margin of the Qinghai-Tibet Plateau, are major part of the Eurasian
Steppe, which is the world's largest grassland (Addison and Greiner, 2016; Wu et al.,
2015). The Chinese grasslands are of great interests in ecological studies, including
grassland biodiversity (Buergi et al., 2015; Yuan et al., 2016), community assembly
(Li and Wu, 2016; Niu et al., 2016), stoichiometry (Wang et al., 2017b), and
ecosystem function (Jing et al., 2015; Mao et al., 2017). Meanwhile, a few studies
have also discussed the origin and dispersal routes of Chinese grassland genus (Favre
et al., 2016; Luo et al., 2016). However, the formation and evolutionary processes of
Chinese grasslands are still rarely studied.
According to paleogeographical climate change, Chinese grasslands might first
emerge in the late of the Tertiary Period because of the global cooling and
aridification (Wu et al., 2015). In addition, Meang and McKenna (1998) found that
perissodactyl-dominant    faunas    of    the    Eocene    were    abruptly    replaced    by



rodent/lagomorph-dominant faunas of the Oligocene in Mongolia Plateau at 33 MaBP,
similar to the European *Grande Coupure*. And they think the turnovers were mainly
driven by global climatic shifts and prominent biotic reorganization from forest to
grassland. Based on the evidence of sporopollen, more studies focused on vegetation
shift and climate change in steppe area of China during the mid-late Holocene
(Mischke et al., 2016; Shen et al., 2008). These studies are of great importance for
revealing the origin and evolution of Chinese grasslands. However, they were mainly
based on a certain history period, and did not explain the expansion routes of Chinese
grasslands. We need more direct evidence to obtain a comprehensive and continuous
history of grassland evolution. Based on currently molecular information, which
providing more accurate and direct evidence for the evolutionary process,
biogeography can be used to deduce the origin and expansion routes of Chinese
grasslands and to explore the effects of important geological historical events (Favre
et al., 2016; Ferreira et al., 2017).

69        Although Chinese grasslands is vast and covers different vegetation types, Stipa

species are dominant through the whole Eurasian steppe, including Chinese
grasslands (Durka et al., 2013; Hamasha et al., 2012). Because of differences in
climate, Stipa species show obvious zonal distribution characteristics in Chinese
grasslands. *Stipa baicalensis*, *Stipa grandis*, and *Stipa krylovii* are constructive
species in the typical grasslands; *Stipa tianschanica* and *Stipa glareosa* are distributed
extensively in the desert steppe; and *Stipa purpurea* has a very extensive distribution
in alpine steppes. In addition, *Stipa basiplumosa* and *Stipa subsessiliflora* are also
distributed in the high mountains of desert areas (Durka et al., 2013; Wu et al., 2015).
Therefore, Stipa species play very important roles in the formation of Chinese
grasslands. The evolutionary processes of Chinese grasslands are closely related to the
evolution of Stipa species. Studies on the history of origin, divergence and expansion
routes of Stipa species are an excellent indicator to reveal the evolution of Chinese
grasslands. Therefore, in this study, Stipa species that were distributed extensively in
different grassland areas including Inner Mongolia, Qinghai, Tibet, and Xinjiang were
collected. The four gene fragments of chloroplasts, *matk, rbcl, trnh-psba,* and *trnl-f,*



were selected to estimate the divergence time tree of Stipa species using BEAST
(Bayesian evolutionary analysis by sampling trees). In addition, the origin, divergence
and expansion routes of Stipa species were reconstructed based on RASP
(Reconstruct Ancestral State in Phylogenies). Finally, the origin and evolutionary
processes of Chinese grasslands were deduced and the effects of paleoclimate and
geological historical events were further explored.

## 2 Materials and Methods

### 2.1 Materials

This study contained 20 species of Stipa species collected from seven different
grasslands area (Fig 1, Table 1). *Helictotrichon schellianum*, *Achnatherum splendens*,
and *Ptilagrostis pelliotii*, which are closely related to Stipa species, were used as the
outgroup (Hamasha et al., 2012). Considering there is certain hybridization between
*Stipa* species(Gonzalo et al., 2012), nuclear-derived markers were not employed, and
the four gene fragments of chlorophyll, *matk*, *rbcl*, *trnh-psba*, and *trnl-f*, were used for
analyses.

### 2.2 Methods

2.2.1 Total DNA extraction, PCR amplification, and sequencing

This study performed total DNA extraction using the Plant Genome Extraction
Kit (Tiangen Biotech,Beijing, China). DNA samples were diluted to 30 ng/μL for
future use(Ramirez et al., 2017). The four selected gene fragments were amplified by
PCR. The universal primers were acquired from previous studies (Durka et al., 2013;
Hamasha et al., 2012; Ren et al., 2011). The PCR reaction volume was 50 μL;
specifically, 2 μL of DNA template, 2 μL each of upstream and downstream primers,
25 μL of 2×PfuUltra Master Mix, and 24 μL ddH$_2$O were combined (Lu et al., 2013).
The *matk* gene amplification was carried out using the following procedure:
pre-denaturation at 94 ℃ for 5 min; 30 cycles of denaturation at 94 ℃ for 30 s,
annealing at 40 ℃ for 30 s, and extension at 72 ℃ for 1 min; and a final extension at
72 ℃ for 10 min. The same amplification procedure was used for the *trnh-psba*
sequence. For the amplification of *trnl-f* and *rbcl*, the annealing temperatures were



changed to 57 ℃ and 52 ℃, respectively, and the other conditions remained
unchanged. The amplification products were subjected to 1% agarose gel
electrophoresis and sent to the Beijing Genomics Institute for sequencing when the
samples were ready.
2.2.2 Sequence comparison
The molecular data obtained from sequencing were assembled using BioEdit and
compared using ClustalW. The sequence length, conserved sites, and GC content were
analyzed using MEGA5.0. These sequence data have been submitted to the GenBank
databases under accession number (see appendices).
2.2.3 Divergence time tree construction using BEAST
The combined sequences of the four gene fragments were used for divergence
time tree construction Therefore, incongruence length difference (ILD) analysis
(heterogeneity test) on the combined gene fragments was required. This analysis was
performed using PAUP (Matthews and Rosenberger, 2008). The results showed that
P>0.01, indicating that these fragments did not have an obvious conflict and could be
used for combined analysis (Rix and Harvey, 2012). In addition, the model selection
software jModeltest (Posada, 2008) was used to calculate the screening of the optimal
model; the selected model was GTR+G. Because there was no suitable Stipa fossil or
gene fragment evolution rate, the average evolution rate of chloroplast gene of
herbaceous plants ($3.46 \times 10^{-9}$ s/s/y) was used to calculate the divergence times. First,
the model was operated for 100 million generations in BEAST (Drummond and
Rambaut, 2007). After the operation was finished, TRACER V1.6 (Rambaut and
Drummond, 2013) was used to look up the ESS value. If the ESS value was larger
than 200, the result was considered reliable. Finally, the final divergence time tree was
checked in FigTree (Rambaut, 2009), which provided the divergence time for Stipa
species.
2.2.4 Reconstruction of historical distribution areas of Stipa species based on RASP
This study used the RASP software to deduce the ancestral distribution areas of
internal nodes in the tree. The Chinese grasslands were divided into seven sections:
(A) eastern Inner Mongolia, (B) central Inner Mongolia, (C) the Helan Mountains, (D)



the Qilian Mountains, (E) Qinghai, (F) Tibet, and (G) Xinjiang. The S-DIVA analysis
used all 100 trees and combined trees in the Bayesian collection. A total of 2500
random trees were selected in the analysis. The maximum number of distributions in
each distribution area was set as two, and the remaining settings were the default ones
(Lu et al., 2013).
**3 Results**
**3.1 Results of sequence feature analysis**
The combined analyses of the four gene fragments showed that the length of the
combined gene fragments was 4093 bp; 2988 bp were conserved sites, 1102 bp were
mutation sites, 227 bp were parsimony-informative sites, and 847 bp were single
information sites. The percentage of G+C in the whole sequence was 37.5%, which
was much lower than the A+T content.
**3.2 Divergence time of Stipa species**
The results showed that the divergence time between Stipa species and the
outgroup (*Achnatherum splendens*, *Helictotrichon schellianum*, and *Ptilagrostis*
*pelliotii*) was 28.42 MaBP (Oligocene period) based on the joint matrix estimate (Fig.
2). The divergence time of Stipa species was 22.10 MaBP (early Miocene period).
There is a explosive rapid radiation of Stipa species around 6.0 MaBP (Fig.2).
**3.3 Analysis of reconstruction of ancestral distribution areas of Stipa species**
**using RASP**
RASP analyses showed that Stipa species originated and diverged from the
outgroup at 28 MaBP (Fig. 3). At node ① (22 MaBP), Stipa species began to
differentiate in central Inner Mongolia. The current distribution pattern of Stipa
species was formed through 16 expansions and 13 isolated divergences, and the
process was more complicated. At node ② (18 MaBP), Stipa species first expanded
to near the Qilian Mountains, Qinghai. Subsequently, at node ③ (13-11 MaBP),
there was a stronger interaction between the top and bottom of mountains in the
Qilian Mountains and other areas in Qinghai. At node ④ (16 MaBP), Stipa species
had already expanded from Qinghai to Tibet; however, no large-area expansion



occurred immediately. Instead, at node ⑥ (11 MaBP-6 MaBP), Stipa species had an
east to west expansion in Tibet. At node ⑤ (12 MaBP), Stipa species expanded
along another route from central Inner Mongolia to the Helan Mountains. At node ⑦
(8 MaBP), the expansion route was from central Inner Mongolia to Xinjiang. Finally,
at node ⑧ (4 MaBP), the expansion route of the Stipa species was from central to
eastern Inner Mongolia.
**3.4 Important influences of paleoclimate and geological historical events on the**
**evolutionary history of Stipa species**
Fig 4 shows that Stipa species underwent three larger expansion and isolation
events. The first larger expansion occurred around 22 MaBP, the second larger
expansion occurred approximately 19-16 MaBP, and the third larger expansion
occurred around 12-10 MaBP. The peak values of isolation events and expansion
events of Stipa species were basically matched. The peak values had certain
relationships with the second uplift of the Qinghai-Tibet Plateau and paleoclimate
change, including East Asian monsoon formation and polar ice cap development.
**4 Discussion**
**4.1 Origin and differentiation of Stipa species**
Mountain building may trigger the origin and radiation of species by providing
vacant niches and habitat alternatives within a short distance (Favre et al., 2016;
Hoorn et al., 2013; Sun et al., 2012), for example in Andes (Hughes and Atchison,
2015), Himalaya-Hengduan Mountains (Luo et al., 2016) and uplift of the
Qinghai-Tibet Plateau (Sun et al., 2012). During the second (23-15 MaBP) and the
third (since 8 MaBP) uplift periods of the Qinghai-Tibet Plateau, many species groups,
including *Nannoglottis* (Liu et al., 2002), Chinese sisorid catfishes (Guo et al., 2006),
*Rheum* (Sun et al., 2012) and *Gentiana* (Favre et al., 2016). Therefore, the uplift of
the Qinghai-Tibet Plateau plays important role in the origin and divergence of many
species.
In this study, the results showed that Stipa species differentiated from the
outgroup and originated at 28 MaBP and began to differentiate at 22 MaBP in central
Inner Mongolia (Fig.3b). It is consistent with a previous study showing that Stipe





species were at least originated in the Miocene or Pliocene through fossil evidence in
North American (Thomasson, 1978). During this period, Himalayan movement had
already uplifted the Qinghai-Tibet Plateau to the height above 2000 m that had a
critical function in the formation of monsoon circulation (Molnar et al., 1993;
Tapponnier et al., 2001). With the continuous uplift and expansion of the plateau,
summer sea surface pressure on the Asian continent increased continuously. With the
presence of a monsoon climate, the humid climate that used to penetrate from Inner
Mongolia to northern Xinjiang no longer existed. This area became particularly arid,
which provided possibilities for the origin and first expansion of Stipa species.
Around 6.0 MaBP, there is a dichotomous relationship among these Stipa species,
and the internodes are consistently short relative to the average tip nodes (Fig.2).
These two results both provide clear indications of explosive rapid radiation in the
past (Hughes and Atchison, 2015; Sun et al., 2012). We speculate that this is the
results of the third uplift of the Qinghai-Tibet Plateau occurred during the 9-2.61
MaBP period (Molnar et al., 1993; Tapponnier et al., 2001). There may be two major
speciation mechanisms that caused the explosive rapid radiation of Stipa species at
that time. On one hand, the first speciation mechanism may be allopatric speciation
(Boucher et al., 2016). Due to the crumpling effect of the uplift of the Qinghai-Tibet
Plateau, the Tian Shan, Qilian, Altyn Tagh, and Kunlun Mountains all had a
large-scale elevation of fault blocks, and many areas that were already elevated
became medium-height mountains with around 4000 m height (Tapponnier et al.,
2001). During this period, geographic isolation was great obvious. As long as there
was obvious geographic isolation, species groups were divided into several small
species groups; because there was no continuous gene flow, different new species
would be generated (Lee and Lin, 2012; Tedesco et al., 2017). On the other hand, for
the area without obvious geographic isolation, such as the Inner Mongolian plateau,
we considered ecological speciation occurred in the presence of gene flow (Shafer and
Wolf, 2013). The uplift of the Qinghai-Tibet Plateau would have caused larger climate
fluctuations in Inner Mongolia Plateau (Tapponnier et al., 2001), thus gradually
generated different small geographic environments. These different small geographic



environments would cause species to occupy different ecological niches, resulting in
natural or sexual selection, which would have caused individuals of the ancestral
species group to undergo phenotype divergence to generate new species (Greve et al.,
2017; Shafer and Wolf, 2013). Therefore, the explosive rapid radiation of Stipa
species around 6.0 MaBP was the result of with and without gene flow driven by
geographic isolation and climate changes in different region. In addition, analysis of
their molecular characteristics showed that the G+C content accounted for 37.5% of
the total sequence length, which was much lower than the A+T content. A and T are
connected and expanded by two hydrogen bonds; therefore, they are more prone to
mutations than G and C (Lee et al., 2016). In addition to external environmental
factors, the molecular characteristics of Stipa species also made the occurrence of
rapid divergence possible.

**4.2 Influence of the Qinghai-Tibet Plateau uplifts and paleoclimate changes on**

**the origin and evolution of the grasslands**

During the developmental process of the whole geological history, various
geological history events continuously occur such as large tectonic movements, the
rise and fall of sea levels, magmatic activities, and volcanic eruptions. Climate
changes on earth are closely associated with these geological history events, whereas
the development of the biosphere is directly influenced by climate and terrain changes
(Guo et al., 2002; Molnar et al., 2010). In China, one of the geological history events
that had great influences on the biosphere were several larger uplift events of the
Qinghai-Tibet Plateau. After the collision between the Indian and European continents,
the Qinghai-Tibet area entered into a whole new developmental stage consisting of
mainly orogenic, fault, and magmatic activities (Tapponnier et al., 2001). There were
three main stages of uplift. The first stage occurred before 30 MaBP (Turner et al.,
1993), the second stage occurred from 23-15 MaBP, and the third stage occurred at 8
MaBP (Molnar et al., 1993). The uplift of the Qinghai-Tibet Plateau caused dramatic
climate changes on earth at that time, and the climate became dry and cold; these
changes were conducive to the origin and evolution of grasslands (Turner et al., 1993;
Wu et al., 2015).





Stipa species started to differentiate at 22 MaBP in central Inner Mongolia;
therefore, during the same period, this area already had a preliminary grassland
landscape. This grassland vegetation structure was also consistent with
rodent/lagomorph-dominant mammal faunas of the Oligocene in Mongolia Plateau
based on fossil evidence (Meng and McKenna, 1998). Afterward, when the expansion
of Stipa species occurred at 16 MaBP, grassland landscapes also emerged successively
in the Qilian Mountains, Qinghai, and western Tibet areas. The major event of
geological history at this stage was the second uplift of the Qinghai-Tibet Plateau. In
addition, factors such as the expansion of rifts in the Asian marginal sea and the
partial shrinkage and disappearance of the eastern extension of the Neo-Tethys Ocean
in Central Asia all resulted in the formation of the Neogene monsoon climate (Huang
et al., 2003; Wang et al., 2003). After it developed at 22 MaBP, the monsoon climate
continuously strengthened and became strongest at 16 MaBP. The strengthening of the
monsoon climate and aridification of the inland areas were synchronous, thereby
causing the further expansion of the arid areas in northwestern China. These events all
provided appropriate environmental conditions for the formation and expansion of the
grasslands (Guo et al., 2002; Wang et al., 2008). The late Miocene period in
geological history occurred approximately 12 MaBP. During this period, the Asian,
and even the global, environment also underwent significant changes. The Arctic area
had a large amount of ice rafts, the polar ice cap developed further, the climate
became drier and colder, an outbreak of $C_4$ plants occurred, and $C_3$ plants decreased
rapidly (Cerling et al., 1997). Based on these events, central Tibet and the ancient
Helan Mountains in Inner Mongolia also exhibited grassland landscapes. During the
period of 9-2.61 MaBP, the third uplift of the Qinghai-Tibet Plateau occurred. The
uplift of the Qilian Mountains, Tian Shan Mountains, Altyn Tagh Mountains, and
Kunlun Mountains made the inland areas more arid (Molnar et al., 1993). During this
period, western Tibet, Xinjiang, and eastern Inner Mongolia also successively
exhibited grassland landscapes. The uplift of the Qinghai-Tibet Plateau and changes in
paleoclimate jointly promoted the origin of grasslands.
**5 Conclusion**





In summary, Stipa species originated at 28 MaBP and they started to differentiate
in central Inner Mongolia at 22 MaBP. Then, Stipa species expanded along four routes.
Based on the expansion route of Stipa species, we deduced that the Chinese
grasslands formed in the following order: central Inner Mongolia > Qilian Mountains >
Qinghai > western Tibet > Helan Mountains > Xinjiang > central Tibet > eastern
Tibet > eastern Inner Mongolia. In addition, the origin and evolution of Stipa species
and Chinese grasslands were accompanied by the uplift of the Qinghai-Tibet Plateau
and paleoclimate change.
**6 Author contributions**
Q.Z. conceived the ideas. Q.Z. and J.C. complied the experiment and ran further
data analysis. Q.Z., C.L. and Y.D. collected study material . Q.Z. and J.C. led the
writing.
**7 Conflict of Interest Statement**
The authors declare that the research was conducted in the absence of any
commercial or financial relationships that could be construed as a potential conflict of
interest.
**8 Acknowledgements:**
We are grateful to professor Frank Yonghong Li and Morigen, for proposing
some valuable suggestions in experimental design. We also grateful professor Cunzhu
Liang for collected some study material. This study was supported by the National
Natural Science Foundation of China (31560180) and China Postdoctoral Science
Foundation.

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






**Tables**

**Table 1** Relevant information about samples used in the experiment. The latin name indicates
Stipa species latin name. The origin indicates the location of Stipa species. The altitude indicates
Stipa species altitude. The latitude and longitude indicates Stipa species latitude and longitude.
The area type indicates the partition type of Stipa species in RASP software.

| Species name | Latin name | Origin | Altitude (m) | Latitude and longitude | Area type |
|---|---|---|---|---|---|
| *Stipa baicalensis* | *Stipa baicalensis* | Hulunbuir, Inner Mongolia | 1650 | 49°20'57"N 120°07'09"E | A |
| *Stipa krylovii* | *Stipa krylovii* | Xilingol League, Inner Mongolia | 930 | 47°51'58"N 115°46'48"E | B |
| *Stipa capillata* | *Stipa capillata* | Jungar Banner, Inner Mongolia | 895 | 39°26'27.7"N 111°09'43.2"E | B |
| *Stipa grandis* | *Stipa grandis* | Xilingol League, Inner Mongolia | 1686 | 44°30'25.9"N 117°21'47.1"E | B |
| *Stipa breviflora* | *Stipa breviflora* | Darhan Muminggan United Banner, Inner Mongolia | 1376 | 41°50'18.1"N 110°13'45.3"E | B |
| *Stipa klemenzii* | *Stipa klemenzii* | Saihan Tala | 1123 | 42°48'49.9"N 112°36'18.4"E | B |
| *Stipa glareosa* | *Stipa glareosa* | Erenhot, Inner Mongolia | 942 | 43°36'42.2"N 111°59'29.9"E | B` |
| *Stipa tianschanica* | *Stipa tianschanica* | Helan Mountains, Inner Mongolia | 1715 | 38°42'49.5"N 105°58'43.7"E | C |





| | | | | | |
|---|---|---|---|---|---|
| *Stipa aliena* | *Stipa aliena* | Qilian Mountains, Qinghai | 3201 | 37°36'38.9"N 101°19'31.3"E | D |
| *Stipa penicillata* | *Stipa penicillata* | Haibei, Qinghai | 3201 | 37°36'43.7"N 101°19'09.0"E | D |
| *Stipa regeliana* | *Stipa regeliana* | Haibei, Qinghai | 3220 | 37°37'04.6"N 101°19'32.3"E | D |
| *Stipa przewalskyi* | *Stipa przewalskyi* | Haixi, Qinghai | 2936 | 37°13'09.5"N 101°32'11.8"E | D |
| *Stipa purpurea* | *Stipa purpurea* | Haibei, Qinghai | 3456 | 37°03'39.0"N 101°41'55.5"E | E |
| *Stipa orientalis* | *Stipa orientalis* | Nagqu, Tibet | 4483 | 31°26'26.6"N 92°01'07.2"E | F |
| *Stipa roborowskyi* | *Stipa roborowskyi* | Tibet | 4805 | 30°29'02.3"N 81°10'19.0"E | F |
| *Stipa subsessiliflora* | *Stipa subsessiliflora* | Tibet | 3444 | 32°57'24.28"N 95°15'46.05"E | F |
| *Stipa capillacea* | *Stipa capillacea* | West Ngamring County, Tibet | 4487 | 29°19'27.27"N 86°58'28.81"E | F |
| *Stipa capillacea* | *Stipa basiplumosa* | Tibet | 4655 | 30°20'09"N 82°54'31.6"E | F |
| *Stipa caucasica* | *Stipa caucasica* | Sai Hubei, Xinjiang | 2170 | 44°54'03.8"N 81°43'20.9"E | G |
| *Stipa sareptana* | *Stipa sareptana* | Xinjiang | 1750 | 49°24'55.1"N 95°26'36.5"E | G |
| *Achnatherum splendens* | *Achnatherum splendens* | Xilamuren, Inner Mongolia | 1607 | 41°20'48.5"N 111°10'17.0"E | B |
| *Ptilagrostis pelliotii* | *Ptilagrostis pelliotii* | Urad Middle Banner, Inner Mongolia | 1518 | 41°53'36.8"N 107°43'32.2"E | B |
| *Helictotrichon* | *Helictotrichon* | Bayanbulak, | 2470 | 42°54'0"N | G |





| *schellianum* | *schellianum* | Xinjiang | 83°43'0"E |
|---|---|---|---|


**Figure Legends**

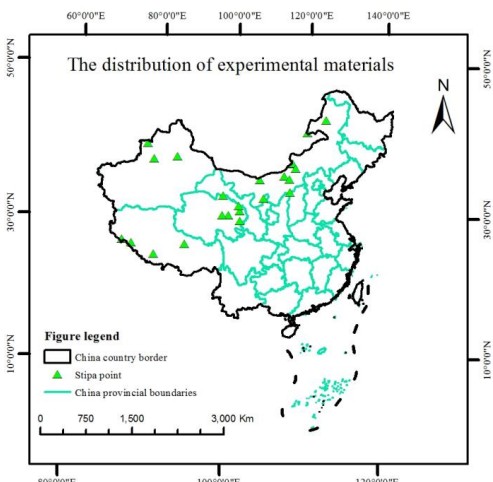


**Figure 1** Distribution of sample material collection points. The green triangle means the
distribution of Stipa species. The green line represents the China provincial boundaries. The black
frame means China country borders.

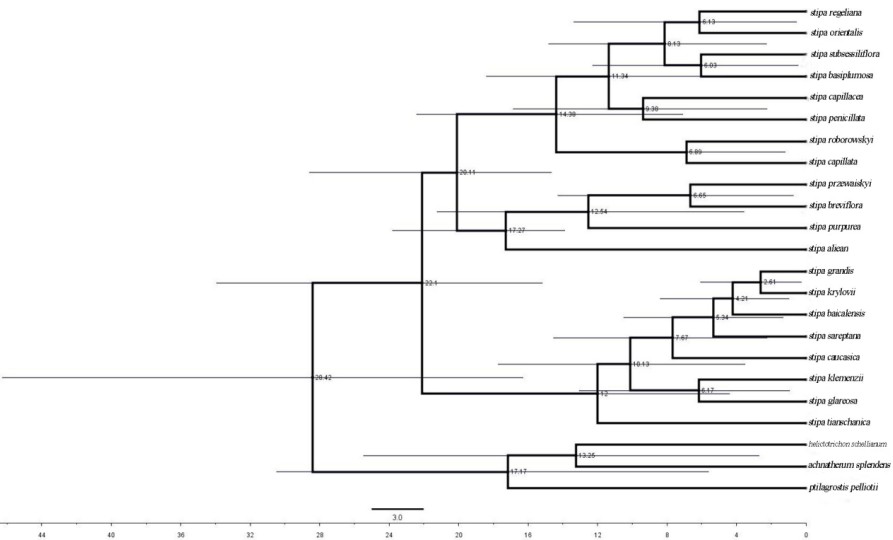


**Figure 2** Divergence time of Stipa species based on BEAST (Units: Ma, million years). The right



side of the picture is the Latin name of the Stipa species. The number of branches is the
divergence time of Stipa species.3.0 means the branch length of divergence time tree. The line
segment represents time scale on the bottom of picture.

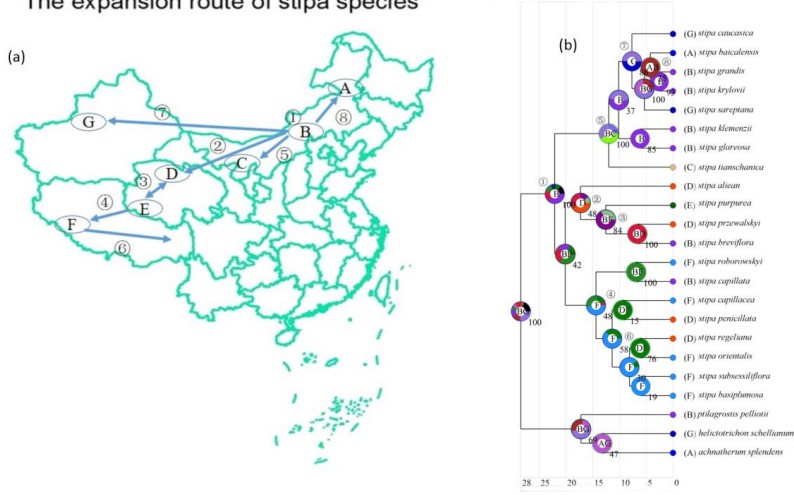



**Figure 3** Analysis of ancestral geographic distribution of Stipa species using RASP. In figure 3(a),
The Chinese grasslands were divided into seven sections: (A) eastern Inner Mongolia, (B) central
Inner Mongolia, (C) the Helan Mountains, (D) the Qilian Mountains, (E) Qinghai, (F) Tibet, and
(G) Xinjiang. The one-way arrows indicate the expansion route calculated using RASP, the
two-way arrow indicates the phenomenon of mutual expansion between two areas. In figure 3(b),
The right side of the picture is the Latin name of the Stipa specie. The letter on each node of the
tree represents the largest possible distribution area of Stipa species in the corresponding time
period. The line segment represents time scale on the bottom of picture. ①-⑧ in Figs 3a and 3b
are landmark nodes of the origin, divergence, and expansion of Stipa species.

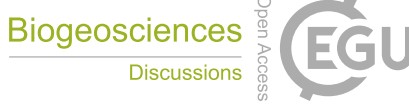



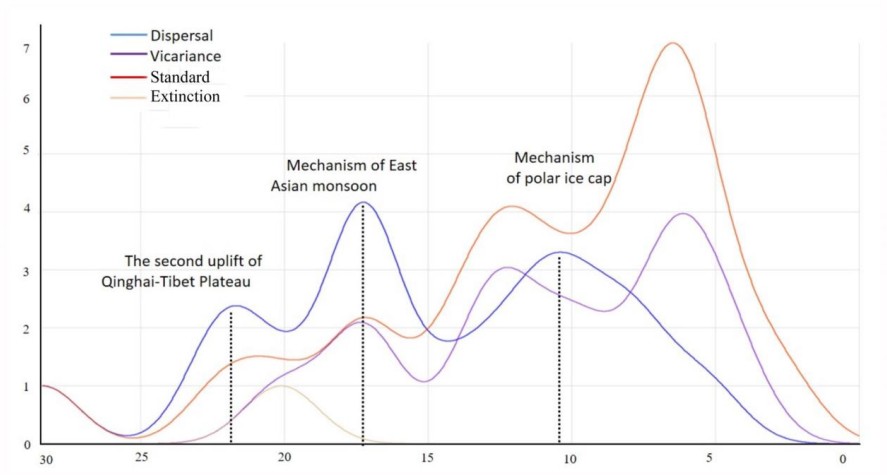

**Figure 4** The time-geological time curve. The population dynamics of Stipa species was described by time abscissas and range ordinates. There are four curves in the graph, and the blue curve represents the expansion trend of Stipa species, the purple curve represents the vicariance trend of Stipa species, the pink curve represents extinction of Stipa species, the red curve is the standard curve.