# Peer review of "Phylogeny of the Stipa and implications for grassland evolution in China: based on biogeographic evidence"

_Biogeosciences, 2018_

## Referee Comment (RC1) · Anonymous Referee #1 · 24 May 2018

This manuscript focuses on an interesting topic - the biogeography and diversification of an ecologically important grass lineage; however, the data presented are insufficient to justify the conclusions that are drawn, and several additional analyses should be done before this manuscript is published in Biogeosciences. I am not convinced that the basic premise of this paper makes sense given that Stipa is a clade with about 350 species that occur on six continents (GrassBase) and only a small number of species from a limited geographic area are represented here. The biogeographic analysis that is presented relies on the assumption that the Chinese Stipa species form a clade and assumes that these lineages have not arrived in China from other regions e.g. Southeast Asia or Europe. For instance it seems possible that some of the Chinese Stipa

might actually have sister species or other close relatives outside of China. There are sequences for more than 75 different species of Stipa available on GenBank and as many of these as possible should be included. With a more complete (or global) phylogeny of Stipa, inferences about dispersal and vicariance events would be much more reliable. I also do not think the data in this manuscript are sufficient to test anything about the diversification rates of Stipa. Again a broader phylogeny that includes Stipa from outside of China would be necessary for such an analysis. Aside from sampling issues, I think that the results of this paper are stated too strongly. There is uncertainty in the phylogentic reconstruction, the inferred divergence dates, and in the biogeographic reconstruction, and very little of this uncertainty is presented or discussed in the current manuscript. Finally, there are grammatical mistakes throughout the manuscript and overall this paper needs to be thoroughly edited for clarity. One of the most common errors is "grasslands" should be plural rather than singular.

Introduction: There should be a section in the introduction with background information about Stipa. How many species are there? What is its distribution outside of China? Maybe also something about the ecology of the clade could be added.

Line 49. See also Liu et al. 2018 "Phylogeny of Orinus (Poaceae), a dominant grass genus on the Qinghai-Tibet Plateau"

Lines 51-57. "Late Tertiary" seems like a misleading description of when Chinese grasslands emerged. "Mid tertiary" would be more appropriate for 33MaBP. Also see Edwards et al. 2010.

Line 73. What are "constructive species?"

Methods: Line 99. Chloroplast not chlorophyll

105. Why is Ramirez et al. 2017 cited related to diluting DNA samples?

Line 134 where is this rate from? Is it possible to get a rate of chloroplast evolution that is specific to grasses?

Line 138-139. The sentence structure makes it sound as though FigTree was used to infer divergence times. Instead "Finally the tree and divergence times were visualized in FigTree."

Lines 145-147. I'm confused whether 100 or 2500 posterior trees were used for this analysis.

Line 148. Setting the maximum number of areas a species can occupy to 2 seems justifiable but needs more context. How many areas do the most widespread Stipa species today occupy?

More details are needed on the biogeographic analysis. How were results summarized across the posterior trees?

What analysis was used to generate Figure 4?

Results: Lines 153-155. This isn't hugely important, but these numbers don't really add up in a way that makes sense. With 2988 conserved and 1102 variable sites there would only be 4090 sites total - what happened to the other three? I'm guessing it has to do with idels or gaps in the alignment? I'm not sure this information is necessary to report, but if you're going to include it, it would make sense to include something about indels as well.

Did the BEAST analysis converge? Were all ESS values >200?

Was the tree well supported? The posterior probability for each node should be added to Figure 2.

Line 160. What are the confidence intervals for these ages?

Line 162. The tree alone does not provide adequate evidence for an "explosive radiation" of Stipa. This could be tested using BAMM (Rabosky et al. 2013) or other diversification methods.

Lines 165-179. This scenario could be plausible if these species form a clade, but there

are other possibilities. Moderate support for other scenarios seem to be indicated by multiple colors in the backbone nodes in Figure 3.

Line 182. What are "larger expansion and isolation events"? How are these different from the expansion and isolation events described in the previous section?

Discussion: How do the age estimates from this study compare to estimates from Poaceae-wide analyses?

Line 200. See Xing and Ree 2017 for a nuanced perspective on this. "Uplift-driven diversification in the Hengduan Mountains, a temperate biodiversity hotspot"

Line 213-4. These lines are confusing because the entire tree is dichotomous. For the second part maybe it should read "the inner branches are short compared to the tip branches?"

Line 216-225. The discussion of rapid radiation in Stipa should either be qualified or removed since it is not explicitly tested in this paper.

Lines 296-301. Because the Chinese grasslands pre-dated Stipa (last section of Discussion), it doesn't seem possible to infer the order of Chinese grassland formation from biogeographic patterns of Stipa.

Figure 3. What are the colors in 3B? I'm confused by what the "largest possible distribution area of Stipa species" means? Largest compared to what? I think it would be clearer to color the map in part A to match the phylogeny in part B. Then the letters and numbers might not be necessary at all? I'm also confused by the fact that part A and the letters on part B seem to tell a very clear story, but the colors of the nodes suggest that there is a lot of uncertainty in the deeper nodes of the tree. That amount of uncertainty would be expected for this kind of an analysis and it seems like it should be reflected more clearly here and in the text of the paper.

Figure 4. How was this graph generated? The colors on this graph aren't very clear—red/pink/purple all look similar to me. Maybe the lines could be thicker or a

green/light blue color could be used instead of one of the other colors.

---

## Referee Comment (RC2) · Anonymous Referee #2 · 19 Jun 2018

The authors use a time calibrated phylogenetic reconstruction of 20 Stipa species to reconstruct the origins and spread of grasslands in China. The premise of this study is based on the observation that several Stipa species are restricted to different regions and the idea that the evolutionary history of these species can serve as a proxy for the history of grassland development. There are several issues with the premise of the study. First, the evolutionary history of one lineage is not enough evidence to draw conclusions about the history of a community. While Stipa is a dominant grass species in these habitats it is not the only grassland species and the authors do not discuss any paleontological evidence to suggest that Stipa has always been a major component of these grasslands. For example, the species of Stipa included in the study may have

evolved after the respective grasslands and subsequently invaded and became domi-
nant. However, if various grassland species in the region were found to have a common
origination time frame then one could conclude that the community began to assemble
at that time. This issue could be resolved by restating the goals of the study to focus
on the evolutionary history of Stipa without the assumption that the history of Stipa is
a good proxy for the evolution of the grasslands that they are found in. A second major
issue is that the researchers only consider 20 species of Stipa in a genus with over
100 species and there is no indication that the 20 species represents a monophyletic
group. Since there are potentially many missing taxa, each with unsampled geograph-
ical distributions, the ancestral area analysis and any inference about dispersal routes
and timing are not reliable. Each of the taxa included in the study could have a sister
species from a different geographical region perhaps from outside the study area. If
so, that would affect both the inference about the pathway of dispersal and the timing
of when speciation events occurred. The authors should investigate the availability of
additional Stipa species on Genbank or other public sequence databases. A third ma-
jor issue relates to the methods used to calibrate the phylogeny. The parameters used
for the BEAST analysis are not clearly stated; however, it appears that the authors
assumed a strict molecular clock with a nucleotide substitution rate based on the "rate
of chloroplast gene of herbaceous plants", although the source of the substitution rate
was not given. There are other studies which report substitution rates in the grass fam-
ily which substantially from the rate used by the authors; however, calibrations based
only on substitution rates are not very reliable unless there is a well-established rate
for the group of organisms. A better method uses fossils to calibrate the stem nodes
of clades to which the fossil is assigned. There are fossils of Stipa, or at least close
relatives of Stipa, as well as other grass fossils which could be used to calibrate the
phylogeny. This would require expanding the phylogeny to include outgroup clades for
which fossils are available. Sequences are available on Genbank that could allow the
authors to do this. For the ancestral area analysis to be meaningful, the phylogeny esti-
mate needs to be well supported. The authors did not clearly report BPP support from

the BEAST analysis nor did they compare the results of their phylogeny estimate with previous phylogeny estimates of Stipa (i.e. Hamasha et al. 2012). Specific Comments Abstract – The abstract is clear – the dates given for grassland formation are very precise some indication of the variance is needed here. Introduction - The grammar needs to be corrected in several places. Lines 27-37: This paragraph is unnecessary since it describes fundamental assumptions that the readers should already be familiar with. Line 38: This paragraph is a better way to start the paper Line 58: "sporopollen" should be "pollen" Lines 58-61: Studies focusing on the Holocene probably are too recent to be important for understanding the origins of grasslands. Line 99: "fragments of chlorophyll" should be "chloroplast fragments" Methods Section 2.2.1 - Good Section 2.2.2 – Combine this section with the previous section. Section 2.2.3 Line 120: Change "assembled" to "aligned". Line 133-134: The average evolution rate of chloroplast gene of herbaceous plants ($3.46 \times 10-9s/s/y$) was used to calculate the divergence times. The resulting 95% HPD of node age estimates is very wide indicating that crown divergence of Stipeae may have occurred between 15-34 million years ago. This wide confidence interval is not adequately discussed by the authors. There is no discussion of how well the topology is supported. There are what appear to be bootstrap results on the RASP analysis but there was no boot strap analysis reported. Lines 133-134 – What are the other parameters of the BEAST run? Lines145-146: "The S-DIVA analysis used all 100 trees and combined trees in the Bayesian collection." - Where did the 100 trees come from? Lines 147-148: "The maximum number of distributions in 148 each distribution area was set as two," . . . Revise this to. . ."The maximum number of ancestral areas was set at two," Lines 155-156: The GC content of the chloroplast is typically much lower that the AT content. Lines 158: No results given for the phylogeny estimation. Discussion The main conclusions can not be supported given the above listed deficiencies in the data. Table 1. Not clear why the species name is repeated. Also, the Table caption is a bit redundant. Only a brief title is needed. Figure 3b: This figure is quite confusing. The colors on the node symbols do not seem to match the tip data and there are more than two ancestral areas represented at internal nodes.

References (mentioned in this review) Hamasha, H. R., von Hagen, K. B., & Röser, M. (2012). Stipa (Poaceae) and allies in the Old World: molecular phylogenetics realigns genus circumscription and gives evidence on the origin of American and Australian lineages. Plant Systematics and Evolution, 298(2), 351-367. Zhong, B., Yonezawa, T., Zhong, Y., & Hasegawa, M. (2009). Episodic evolution and adaptation of chloroplast genomes in ancestral grasses. PLoS One, 4(4), e5297.

---

## Referee Comment (RC3) · Anonymous Referee #3 · 4 Jul 2018

General comments In this manuscript, the authors present an interesting analysis of grassland evolution and biogeography in relation to the Cenozoic history of China. Focusing on Stipa grass species, they 1) generate a time-calibrated phylogeny to estimate divergence times, 2) reconstruct ancestral geographic distributions, and 2) discuss ways in which landscape and climate events may have contributed to geographic expansions and speciation events. In general, this is an interesting question and the approaches appear sound; however, some questions remain about specific methodological decisions and the overall validity of the findings. Furthermore, more caution in inferring the biogeographic history of the Stipa group is encouraged so as not to overstate the study's findings. For example, what are the assumptions that go into the

biogeographic model and what are the errors and/or uncertainties involved in the timing of divergence and geologic/climatic events? Finally, some changes to the figures and manuscript text are recommended for clarity's sake. Please find below suggested line and figure edits.

Introduction Line 32-35: There are a few confusing things in this statement. First of all, one cannot infer biogeographic processes based on molecular clocks/genetic information. A step - ancestral state reconstruction for geography, which the authors perform - provides this information, not the molecular clock or the genetic information itself. Next, what does isolation of organisms at a molecular level mean and how does one assess the importance of geologic events. This sentence would benefit from some editing and conceptual clarification.

Line 36: It is too strong to say that divergence times can be "basically confirmed" – they are inferred or estimated, based on some model of evolution and assumptions about molecular clocks.

Line 51: Suggest change of phrasing and clarification: "According to In relation to paleogeographical climate change..." What does paleogeographic refer to? There is nothing geographic in the statement being made here. Regional climate change in China?

Line 63-67: The rationale for this study is fairly clear, but I would suggest the authors moderate their language, especially with the use of "direct" evidence. Phylogenies and biogeographic models are built on a series of assumptions that enable us to infer the history of grassland expansion in relation to climate and landscape events; although these are often reasonable assumptions, this does not equate direct evidence. Direct evidence would comprise temporal and geographic series of grass fossils/biomarkers/etc., which is beyond the scope of this paper.

Lines 69-80: I am not familiar with the composition of grassland ecosystems in China – what do the authors mean, for example, by "constructive species in the typical grasslands"? And when the authors describe Stipa species as dominant, what does this mean (abundance, diversity)? For readers unfamiliar with Chinese grasslands or Stipa more generally, what is a "typical" grassland (line 74) and what is the broader context for Stipa evolution/biogeography? What is the basic for the statement that "the evolutionary processes of Chinese grasslands are closely related to the evolution of Stipa species"? Can the authors describe more about regional grassland communities? Otherwise, Stipa species are just a case study of one group of grasses, rather than a proxy for all grasslands.

Lines 86-88: Please include references for BEAST and RASP.

Materials and Methods Line 139: Can you clarify how FigTree provided the divergence times for Stipa species. I was only aware that this program helped visualize trees.

Lines 147-148: Why did you assign a maximum number of distribution areas as 2? Please explain your rationale – is that the maximum number of areas any given extant species occurs in?

Results Line 161: What is meant by "the divergence time of Stipa species" in this sentence – I know what you mean when looking at the figure, but in the text it is unclear that this refers to the basal-most split in the studied clade. Since Stipa is a large, widely distributed group, what does this mean in the context of the whole group?

Line 162: I'm curious as to why the authors did not test for radiations/diversification rate shifts directly. If the primary evidence for an "explosive rapid radiation" is from visual inspection of the tree, I am somewhat unconvinced. It would be better to do a formal test for rate variation, such as using BAMM (Rabosky 2014 in PLOS One). If this kind of test is beyond the scope of the paper, I suggest reframing as an inference/hypothesis that could explicitly tested at a later date.

Line 168: I'm curious what the authors mean by isolated divergences – are they referring to a vicariance model? Can the authors clarify, and justify how they determined an

event to be an isolated divergence event?

Line 171: It is unclear what the authors are referring to – stronger interaction than what? What kind of interaction? This brings up another point – for readers unfamiliar with the landscape of the 7 regions, the topographic setting may be unknown and thus hard to know what the strength of the elevational gradient from the "top and bottom" of the mountains means. Please provide a little more information about the regions (perhaps earlier in the Introduction?).

Line 182: Can the authors please explain how Figure 4 was generated? Does RASP provide frequency estimates of event types, and from there, authors generated what looks like a kernel density plot? In Figure 4, there is a "Standard" and "Extinction" line; however, these are not mentioned in the text. Can the authors please elaborate? Furthermore, the colors in the figure do not match the legend provided, so it is difficult to align what is in the text with the figure. Finally, the three geologic events illustrated in this figure seem to occur instantaneously, when in reality these events likely took >1myr. Instead of drawing a single line, I suggest providing the age range of geologic events (similar to how it is presented in the text as a range and not a single date).

Line 185-187: Can the author please clarify what is meant by "isolation events" – is this vicariance, founder event? Seems like the authors mean vicariance, but it would be helpful to be as clear as possible, and for the terminology in the manuscript to match that of the figures. Furthermore, I'm not sure that I see how the peak frequency (I'm assuming this is frequency, although the y-axis is not labeled) in dispersal and vicariance are "basically matched" in time - can the authors conduct a statistical test to verify this?

Discussion Line 195-198: This is an incomplete sentence, please rewrite.

Lines 212: Please include a reference for the changing climate conditions.

Lines 214: I don't see this (short internodes) very clearly; again, this could be formally

tested. Until it is tested, I am unconvinced that this is a clear indication of an explosive, rapid radiation, especially given the large error bars on divergence age estimates. I recommend that the authors moderate their statement – perhaps point to a suggestion of a radiation, but that this remains untested.

Lines 221: There are a few qualitative statements here that I think can be removed: "Due to the crumpling effect of the uplift of the Qinghai-Tibet Plateau, the Tian Shan, Quilian, Altyn Tagh, and Kunlun Mountains all had large-scale elevation of fault-blocks, and many areas that were already elevated became medium-height mountains with around 4000m height."

Line 225: What do the authors mean here? I suggest rephrasing to say that geographic isolation was likely. "Obvious" here in and elsewhere in this paragraph is kind-of a loaded term, and I suggest avoiding it.

Line 230/235: Is there evidence, other than the assumed absence of physical barrier, for ecological and/or sexual speciation? Ancestral state reconstruction of geographic areas helps inform geography of speciation, but not necessarily mode of speciation.

Line 237-239: This statement is too strong, better to be cautious and use language, such as we infer x or evidence supports y... I don't think we can know this definitively, even with a formal test for rate shifts.

Line 240-244: This is an interesting idea. However, I am curious, if Stipa species have a high A+T content and A+T bonds are more prone to mutations, does this imply that the average evolutionary rate of herbaceous plants may be an underestimate of rates for the Stipa group? If so, how might that affect your findings? In general, a more developed discussion of the assumptions that went into the BEAST and RASP analyses would be good, as well as an explanation for the wide error bars on the reconstructed phylogeny.

Line 246-247: It is a little unclear what the division between discussion sections 4.1 and

4.2 is, since geologic and climate history is brought up in 4.1 in relation the divergence dates and geographic expansions. This is up to the authors, but perhaps it would better serve the reader to include some of the background information about the landscape history of the Tibetan Plateau and regions of the study in the Introduction of the paper. Then, the authors could more freely discuss this history throughout the manuscript's discussion.

Line 248: I suggest removing the line "During the developmental process of the whole geological history" since it is a little unclear what this refers to (e.g., the scope of geological history is far greater in space and time than what is explored in this study).

Line 266-268: Can the authors elaborate on how Oligocene faunal turnover (to a rodent-lagomorph dominated fauna) supports their inferred Miocene Stipa expansion? It is still somewhat unclear, to me, how extensive Chinese grasslands were prior to divergence and expansion of the Stipa and/or whether Stipa are a major player in history of grassland expansion. Or, if they are an interesting group to study because of their dominance (?) today and history in relation to more recent (e.g., Miocene) geologic/climate events. I think this remains unclear throughout the manuscript – for example, in Lines 291-292, the authors surmise that the uplift of the Qinghai-Tibet Plateau and climate changes promoted the origin of grasslands, which appears to contradict an earlier origin inferred from faunal turnover and mentioned in Line 266. This confusion might be cleared up by clarifying early on the current state of knowledge (based on fossil evidence, other non-Stipa groups, etc.) and how Stipa specifically contributes to the grassland story in China – e.g., does it signal grassland expansion?

Line 284: Suggest replacing "outbreak" with expansion, shift in ecological dominance, etc. . .

Figures: In general, the text in the accompanying figures is small and difficult to read. Is it possible to enlarge the figures and figure text?

Figure 2: There are very wide error bars on divergence times; this should be mentioned

in the results and should be discussed in detail in the results and/or discussion. What contributes to wide error on divergence age estimates and how does this influence your interpretations of evolutionary processes? Stipa should also be capitalized in the Latin names.

Figure 3: Can you make this figure larger? It is difficult to read as is, especially the ancestral states and landmark nodes on the phylogeny. Furthermore, the colors on the phylogeny seem to correspond with different regions. Can you color code the different regions on the map as well? Provinces appear to contain multiple biogeographic regions, so it is difficult to tell where the region boundaries are. Not necessary, but it might also help get the authors' message across if another panel is included with terrain, so that the readers can know where mountains ranges exist, etc. in relation to the biogeographic regions and inferred dispersal routes.

Figure 4: Please add y-axis and x-axis labels to this figure, and more detail as to how this figure was constructed. Are we looking at output from the RASP analysis? In addition, please change the colors of the curves to match those of the figure legend (for example, I cannot tell which curve is the extinction and which is the standard). In the figure caption, what does a time-geological time curve mean? Do the authors simply mean event curves over geologic time from 30 Ma to present? And, is it necessary to use the terminology "time abscissas and range ordinates"? I think this is confusing, when, I believe, the authors are just describing x and y coordinates.

Frequent typos: e.g., missing spaces between word and reference, missing punctuation; inconsistent pluralization of grasslands, area, etc.; "stipa" is lower case in the figures; comparative adjectives used without a comparison noun (e.g., lines 182-183 – "larger" should be "large" or the authors should state what the expansion is larger than); "MaBP" can just be "Ma"

---

## Author Comment (AC1) · 4 Jul 2018

July 4, 2018

Prof. Christopher Still

Department of Forest Ecosystems and Society, Oregon State University

321 Richardson Hall, Corvallis OR 97331-5752

Dear Prof. Christopher Still,

We would like to thank you for the opportunity to discuss our manuscript ID bg-2018-

140 entitled 'Phylogeny of the Stipa and implications for grassland evolution in China: based on biogeographic evidence'. We are grateful to you and the two reviewers for their constructive comments and thoughtful suggestions that are very helpful in improving significantly the quality of our manuscript. We have analyzed all the comments carefully. All major replies are described in detail point-to-point. Please let us know should you have any questions regarding the manuscript. We are looking forward to hearing from you.

Sincerely yours,

Qing Zhang

School of Ecology and Environment, Inner Mongolia University,

No. 235 University West Road. Hohhot, 010021, China.

Tel: +86-471-4992735

Fax: +86-471-4991656

Email: qzhang82@163.com

Response to bg-2018-140 – RC1:

Issue 1. The biogeographic analysis that is presented relies on the assumption that the Chinese Stipa species form a clade and assumes that these lineages have not arrived in China from other regions e.g. Southeast Asia or Europe. For instance it seems possible that some of the Chinese Stipa might actually have sister species or other close relatives outside of China. There are sequences for more than 75 different species of Stipa available on GenBank and as many of these as possible should be included. With a more complete (or global) phylogeny of Stipa, inferences about dispersal and vicariance events would be much more reliable. I also do not think the data in this manuscript are sufficient to test anything about the diversification rates of Stipa. Again a broader phylogeny that includes Stipa from outside of China would be necessary for

such an analysis.

Response: We appreciate for the constructive suggestion and agree with the point. We will investigate the availability of additional Stipa species (global) on GenBank and other public sequence databases and conduct analysis about phylogeny, dispersal and vicariance events.

Issue 2. Aside from sampling issues, I think that the results of this paper are stated too strongly. There is uncertainty in the phylogentic reconstruction, the inferred divergence dates, and in the biogeographic reconstruction, and very little of this uncertainty is presented or discussed in the current manuscript.

Response: We agree with the point. We will add the corresponding uncertainty discussion of phylogenetic reconstruction, the inferred divergence dates, and the biogeographic reconstruction.

Issue 3. Finally, there are grammatical mistakes throughout the manuscript and overall this paper needs to be thoroughly edited for clarity. One of the most common errors is "grasslands" should be plural rather than singular.

Response: We apologize for the grammatical mistakes. We will ask for a native English speaker to check the revised manuscript.

Issue 4. There should be a section in the introduction with background information about Stipa. How many species are there? What is its distribution outside of China? Maybe also something about the ecology of the clade could be added.

Response: We appreciate the suggestion and agree with it. We will review more literature and add an overview of Stipa species numbers, distribution and clade.

Issue 5. Line 49. See also Liu et al. 2018 "Phylogeny of Orinus (Poaceae), a dominant grass genus on the Qinghai-Tibet Plateau"

Response: Thanks to the reviewer for recommending this article. This article has

provided a great help for us to revise our manuscript.

Issue 6. Lines 51-57. "Late Tertiary" seems like a misleading description of when Chinesegrasslands emerged. "Mid tertiary" would be more appropriate for 33MaBP. Also seeEdwards et al. 2010.

Response: We also apologize for the confusion and agree with the point. It seems more appropriate that Chinese grasslands emerged about 33 MaBP during the Mid tertiary.

Issue 7. Line 73. What are "constructive species?"

Response: Constructive species is also called edificato, or edificator species. It is the most dominant species in a community, and also plays a significant control role in community structure and function.

Issue 8. Line 99. Chloroplast not chlorophyll

Response: We apologize for the mistake. We will modify it as the suggestion.

Issue 9. 105. Why is Ramirez et al. 2017 cited related to diluting DNA samples?

Response: We also apologize for the confusion. We will change it to the appropriate literature.

Issue 10. Line 134 where is this rate from? Is it possible to get a rate of chloroplast evolution that is specific to grasses?

Response: We thank the reviewer for the valuable suggestion. We found it is feasible to get a rate of chloroplast evolution of Stipeae to calculate the divergence times of Stipa species from this literature (Romaschenko et al. 2014).

Issue 11. Line 138-139. The sentence structure makes it sound as though FigTree was used to infer divergence times. Instead "Finally the tree and divergence times were visualized in FigTree."

Response: We agree with the reviewer and will revise it as suggestion.

Issue 12. Lines 145-147. I'm confused whether 100 or 2500 posterior trees were used for this analysis.

Response: We apologize for the confusion. The number of 100 should be 10000. The S-DIVA analysis used 2500 random trees which were selected in the Bayesian analysis with a total of 10000 trees.

Issue 13. Line 148. Setting the maximum number of areas a species can occupy to 2 seems justifiable but needs more context. How many areas do the most widespread Stipa species today occupy?

Response: Stipa krylovii is the most widespread species, mainly distributes in Inner Mongolia, Xinjiang, Loess Plateau.

Issue 14. More details are needed on the biogeographic analysis. How were results summarized across the posterior trees?

Response: We apologize for the confusion. We will add more details of biogeographic analysis on revised manuscript.

Issue 15. What analysis was used to generate Figure 4?

Response: We also apologize for the confusion. It is generated by BEAST software. We will add the details in the revised manuscript.

Issue 16. Lines 153-155. This isn't hugely important, but these numbers don't really add up in a way that makes sense. With 2988 conserved and 1102 variable sites therewould only be 4090 sites total - what happened to the other three? I'm guessing it has to do with idels or gaps in the alignment? I'm not sure this information is necessary to report, but if you're going to include it, it would make sense to include something aboutindels as well.

Response: We also apologize for the confusion. We only showed the length of the

combined four gene fragments. We will add corresponding information of each gene fragment.

Issue 17. Did the BEAST analysis converge? Were all ESS values >200?

Response: The BEAST analysis was converge, and all ESS values were greater than 200.

Issue 18. Was the tree well supported? The posterior probability for each node should be added to Figure 2.

Response: Yes. The tree was well supported. We will add posterior probability for each node to Figure 2 in the revised manuscript.

Issue 19. Line 160. What are the confidence intervals for these ages?

Response: Agreed. We will add all age with 95% highest posterior density in the revised manuscript.

Issue 20. Line 162. The tree alone does not provide adequate evidence for an "explosive radiation" of Stipa. This could be tested using BAMM (Rabosky et al. 2013) or other diversification methods.

Response: We thank the review for the constructive suggestion. We will add the BAMM analysis to verify explosive radiation according to the literature(Rabosky et al. 2013).

Issue 21. Lines 165-179. This scenario could be plausible if these species form a clade, but there are other possibilities. Moderate support for other scenarios seem to be indicated by multiple colors in the backbone nodes in Figure 3.

Response: Thanks to reviewer for the value suggestion. Because we will add additional Stipa species sequence database from GenBank, we will explore other scenarios based on multiple colors in the backbone nodes in Figure 3.

Issue 22. Line 182. What are "larger expansion and isolation events"? How are these

different from the expansion and isolation events described in the previous section?

Response: We also apologize for the unclear description. There is no difference between "larger expansion and isolation events" and "expansion and isolation events". We will unified these two the same as "expansion and isolation events".

Issue 23. How do the age estimates from this study compare to estimates from Poaceae-wide analyses?

Response: We appreciate the valuable comment. We will consult the appropriate literature to explore the age estimates compare to Poaceae.

Issue 24. Line 200. See Xing and Ree 2017 for a nuanced perspective on this. "Uplift-driven diversification in the Hengduan Mountains, a temperate biodiversity hotspot"

Response: We thanks the reviewer for recommending this article. This article has provided a great help for us to revise our manuscript.

Issue 25. Line 213-4. These lines are confusing because the entire tree is dichotomous. For the second part maybe it should read "the inner branches are short compared to the tip branches?"

Response: We apologize for the confusion. We will revise it as suggestion "the inner branches are short compared to the tip branches".

Issue 26. Line 216-225. The discussion of rapid radiation in Stipa should either be qualified or removed since it is not explicitly tested in this paper.

Response: As the reply to comment 20, we will add the BAMM analysis to verify explosive radiation according to the literature(Rabosky et al. 2013). Based on the result of BAMM, we will conduct the discussion.

Issue 27. Lines 296-301. Because the Chinese grasslands pre-dated Stipa (last section of Discussion), it doesn't seem possible to infer the order of Chinese grassland formation from biogeographic patterns of Stipa.

Response: We are grateful for the constructive suggestion and agree with this point. We will delete all the content about the order of Chinese grasslands formation.

Issue 28. Figure 3. What are the colors in 3B? I'm confused by what the "largest possible distribution area of Stipa species" means? Largest compared to what? I think it would be clearer to color the map in part A to match the phylogeny in part B. Then the letters and numbers might not be necessary at all? I'm also confused by the fact that part A and the letters on part B seem to tell a very clear story, but the colors of the nodes suggest that there is a lot of uncertainty in the deeper nodes of the tree. That amount of uncertainty would be expected for this kind of an analysis and it seems like it should be reflected more clearly here and in the text of the paper.

Response: We apologize for the confusion. In Figure 3B, each color refers to different distribution area of Stipa species. Based on RASP, we explored the ancestral distribution areas of Stipa species. Then, on the circle node, the ratio of each color represented the proportion of the ancestor distribution area.

Issue 29 Figure 4. How was this graph generated? The colors on this graph aren't very Clear. A red/pink/purple all look similar to me. Maybe the lines could be thicker or a green/light blue color could be used instead of one of the other colors.

Response: Thanks for the useful suggestion. We will modify the Figure 4 as the suggestion.

References:

Rabosky DL, Santini F, Eastman J et al. 2013. Rates of speciation and morphological evolution are correlated across the largest vertebrate radiation. Nature Communications, 4

Romaschenko K, Garciajacas N, Peterson PM et al. 2014. Miocene-Pliocene speciation, introgression, and migration of Patis and Ptilagrostis (Poaceae: Stipeae). Molecular Phylogenetics & Evolution, 70(1):244-259.

Please also note the supplement to this comment:
https://www.biogeosciences-discuss.net/bg-2018-140/bg-2018-140-AC1-
supplement.pdf

———————————————————

---

## Author Comment (AC2) · 4 Jul 2018

July 4, 2018

Prof. Christopher Still

Department of Forest Ecosystems and Society, Oregon State University

321 Richardson Hall, Corvallis OR 97331-5752

Dear Prof. Christopher Still,

We would like to thank you for the opportunity to discuss our manuscript ID bg-2018-

140 entitled 'Phylogeny of the Stipa and implications for grassland evolution in China: based on biogeographic evidence'. We are grateful to you and the two reviewers for their constructive comments and thoughtful suggestions that are very helpful in improving significantly the quality of our manuscript. We have analyzed all the comments carefully. All major replies are described in detail point-to-point. Please let us know should you have any questions regarding the manuscript. We are looking forward to hearing from you.

Sincerely yours,

Qing Zhang

School of Ecology and Environment, Inner Mongolia University,

No. 235 University West Road. Hohhot, 010021, China.

Tel: +86-471-4992735

Fax: +86-471-4991656

Email: qzhang82@163.com

Response to bg-2018-140 – RC2:

Issue 1. First, the evolutionary history of one lineage is not enough evidence to draw conclusions about the history of a community. While Stipa is a dominant grass species in these habitats it is not the only grassland species and the authors do not discuss any paleontological evidence to suggest that Stipa has always been a major component of these grasslands. For example, the species of Stipa included in the study may have evolved after the respective grasslands and subsequently invaded and became dominant. However, if various grassland species in the region were found to have a common origination time frame then one could conclude that the community began to assemble at that time. This issue could be resolved by restating the goals of the study to focus on the evolutionary history of Stipa without the assumption that the history of

Stipa is a good proxy for the evolution of the grasslands that they are found in.

Response: We appreciate the constructive comment and agree with it. We will delete all the content of grassland evolution indicated by the evolution of Stipa species.

Issue 2. A second major issue is that the researchers only consider 20 species of Stipa in a genus with over 100 species and there is no indication that the 20 species represents a monophyletic group. Since there are potentially many missing taxa, each with unsampled geographical distributions, the ancestral area analysis and any inference about dispersal routes and timing are not reliable. Each of the taxa included in the study could have a sister species from a different geographical region perhaps from outside the study area. If so, that would affect both the inference about the pathway of dispersal and the timing of when speciation events occurred. The authors should investigate the availability of additional Stipa species on Genbank or other public sequence databases.

Response: We agree with the reviewer and are grateful for the valuable suggestion. We will investigate the availability of additional Stipa species (global) on GenBank and other public sequence databases and conduct analysis about phylogeny, dispersal and vicariance events.

Issue 3. A third major issue relates to the methods used to calibrate the phylogeny. The parameters used for the BEAST analysis are not clearly stated; however, it appears that the authors assumed a strict molecular clock with a nucleotide substitution rate based on the "rate of chloroplast gene of herbaceous plants", although the source of the substitution rate was not given. There are other studies which report substitution rates in the grass family which substantially from the rate used by the authors; however, calibrations based only on substitution rates are not very reliable unless there is a well-established rate for the group of organisms. A better method uses fossils to calibrate the stem nodes of clades to which the fossil is assigned. There are fossils of Stipa, or at least close relatives of Stipa, as well as other grass fossils which could be used

to calibrate the phylogeny. This would require expanding the phylogeny to include outgroup clades for which fossils are available. Sequences are available on Genbank that could allow theauthors to do this.

Response: We thank the reviewer for the valuable suggestion. We found it is feasible to get a rate of chloroplast evolution of Stipeae to calculate the divergence times of Stipa species from this literature (Romaschenko et al. 2014). Meanwhile, we also will try to consult fossils of Stipa or close species and corresponding available sequences date to further calibrate the Phylogeny.

Issue 4. For the ancestral area analysis to be meaningful, the phylogeny estimate needs to be well supported. The authors did not clearly report BPP support from the BEAST analysis nor did they compare the results of their phylogeny estimate with previous phylogeny estimates of Stipa (i.e. Hamasha et al. 2012).

Response: We thanks for the suggestion. Hamasha et al (2012) studied phylogeny of 109 Stipa species from Eurasia, Americas and Australia. It is very useful for us to further discuss the phylogeny of Stipa in our manuscript.

Issue 5. The abstract is clear – the dates given for grassland formation are very precise some indication of the variance is needed here.

Response: Agreed. We will add all age with 95% highest posterior density in the revised manuscript.

Issue 6. Lines 27-37: This paragraph is unnecessary since it describes fundamental assumptions that the readers should already be familiar with. Line 38: This paragraph is a better way to start the paper

Response: We agree with the reviewer. We will revise it as the suggestion.

Issue 7. Line 58: "sporopollen" should be "pollen"

Response: We apologize for the mistake. We will modify it as the comment.

Issue 8. Lines 58-61: Studies focusing on the Holocene probably are too recent to be important for understanding the origins of grasslands.

Response: We appreciate the comment and agree with the point. We will revise it as the suggestion.

Issue 9. Line 99: "fragments of chlorophyll" should be "chloroplast fragments"

Response: Agreed. We will change "fragments of chlorophyll" to "chloroplast fragments".

Issue 10. Section 2.2.1 - Good Section 2.2.2 – Combine this section with the previous section.

Response: Agreed. We will revise it as the comment.

Issue 11. Section 2.2.3 Line 120: Change "assembled" to "aligned".

Response: Agreed. We will revise "assembled" to "aligned".

Issue 12. Line 133-134: The average evolution rate of chloroplast gene of herbaceous plants (3.4610-9s/s/y) was used to calculate the divergence times. The resulting 95% HPD of node age estimates is very wide indicating that crown divergence of Stipeae may have occurred between 15-34 million years ago. This wide confidence interval is not adequately discussed by the authors. There is no discussion of how well the topology is supported. There are what appear to be bootstrap results on the RASP analysis but there was no boot strap analysis reported.

Response: As reply to comment 3, We will adopt a rate of chloroplast evolution of Stipeae to calculate the divergence times of Stipa species from this literature (Romaschenko et al. 2014). Meanwhile, we also will try to consult fossils of Stipa or close species and corresponding available sequences date to further calibrate the phylogeny.

Issue 13. Lines 133-134 –What are the other parameters of the BEAST run?

Response: The evolution rate of chloroplast gene was the crucial parameter in RASP analysis, and others parameters were basic.

Issue 14. Lines145-146: "The S-DIVA analysis used all 100 trees and combined trees in the Bayesian collection." - Where did the 100 trees come from?

Response: We apologize for the confusion. The number of 100 should be 10000. The S-DIVA analysis used 2500 random trees which were selected in the Bayesian analysis with a total of 10000 trees.

Issue 15. Lines 147-148: "The maximum number of distributions in 148 each distribution area was set as two," : : : Revise this to: : :"The maximum number of ancestral areas was set at two,"

Response: Agreed. We will revise this sentence as the suggestion.

Issue 16. Lines 155-156: The GC content of the chloroplast is typically much lower that the AT content.

Response: Thanks. We will revise the sentence as the comment "The GC content of the chloroplast is typically much lower that the AT content."

Issue 17. Lines 158: No results given for the phylogeny estimation.

Response: We apologize for the confusion. The credibility of phylogeny is determined by the ESS value. If the ESS value is greater than 200, the result is credible. We will add some information of ESS values in the revised manuscript.

Issue 18. Table 1. Not clear why the species name is repeated. Also, the Table caption is a bit redundant. Only a brief title is needed.

Response: We thank the review for the valuable suggestion. We will only retain species names, and named Table 1 as "Relevant information about all Stipa species." in the revised manuscript.

Issue 19. Figure 3b: This figure is quite confusing. The colors on the node symbols do not seem to match the tip data and there are more than two ancestral areas represented at internal nodes.

Response: We apologize for the confusion. In Figure 3B, each color refers to different distribution area of Stipa species. Based on RASP, we explored the ancestral distribution areas of Stipa species. Then, on the circle node, the ratio of each color represented the proportion of the ancestor distribution area.

Issue 20. References (mentioned in this review) Hamasha, H. R., von Hagen, K. B., & Röser, M.(2012). Stipa (Poaceae) and allies in the Old World: molecular phylogenetics realigns genus circumscription and gives evidence on the origin of American and Australian lineages. Plant Systematics and Evolution, 298(2), 351-367. Zhong, B., Yonezawa, T.,Zhong, Y., & Hasegawa, M. (2009). Episodic evolution and adaptation of chloroplast genomes in ancestral grasses. PLoS One, 4(4), e5297.

Response: Thanks to the reviewer for recommending these articles. These articles have provided a great help for us to revise our manuscript.

References

Hamasha HR, von Hagen KB, Roeser M. 2012. Stipa (Poaceae) and allies in the Old World: molecular phylogenetics realigns genus circumscription and gives evidence on the origin of American and Australian lineages. Plant Syst. Evol., 298(2):351-367.

Romaschenko K, Garciajacas N, Peterson PM et al. 2014. Miocene-Pliocene speciation, introgression, and migration of Patis and Ptilagrostis (Poaceae: Stipeae). Molecular Phylogenetics & Evolution, 70(1):244-259.

Please also note the supplement to this comment:
https://www.biogeosciences-discuss.net/bg-2018-140/bg-2018-140-AC2-supplement.pdf

---

## Author Comment (AC3) · 7 Jul 2018

July 7, 2018

Prof. Christopher Still

Department of Forest Ecosystems and Society, Oregon State University

321 Richardson Hall, Corvallis OR 97331-5752

Dear Prof. Christopher Still,

We would like to thank you for the opportunity to discuss our manuscript ID bg-2018-

140 entitled 'Phylogeny of the Stipa and implications for grassland evolution in China: based on biogeographic evidence'. We are grateful to you and the third reviewer for constructive comments and thoughtful suggestions that are very helpful in improving significantly the quality of our manuscript. We have analyzed all the comments carefully. All major replies are described in detail point-to-point. Please let us know should you have any questions regarding the manuscript. We are looking forward to hearing from you.

Sincerely yours,

Qing Zhang

School of Ecology and Environment, Inner Mongolia University,

No. 235 University West Road. Hohhot, 010021, China.

Tel: +86-471-4992735

Fax: +86-471-4991656

Email: qzhang82@163.com

Response to bg-2018-140 – RC3:

Issue 1. Line 32-35: There are a few confusing things in this statement. First of all, one cannot infer biogeographic processes based on molecular clocks/genetic information. A step - ancestral state reconstruction for geography, which the authors perform - provides this information, not the molecular clock or the genetic information itself. Next, what does isolation of organisms at a molecular level mean and how does one assess the importance of geologic events. This sentence would benefit from some editing and conceptual clarification.

Response: We apologize for the confusion and agree with the reviewer for the point. We will modify this sentence based on clear concept and reasonable English style.

Issue 2. Line 36: It is too strong to say that divergence times can be "basically confirmed" – they are inferred or estimated, based on some model of evolution and assumptions about molecular clocks.

Response: Agreed. We will revise it as the suggestion "divergence times can be basically inferred".

Issue 3. Line 51: Suggest change of phrasing and clarification: "According to In relation to paleogeographical climate change: : :" What does paleogeographic refer to? There is nothing geographic in the statement being made here. Regional climate change in China?

Response: We agree with the point and will revise it as "According to in relation to paleogeographical climate change, Chinese grasslands might first emerge in the late of the Tertiary Period. At that time, because of the uplift of the Qinghai-Tibet Plateau, the Himalayas Mountains blocked the warm moist air mass coming from the Indian Ocean, and the climate of the Mongolian Plateau became colder and drier. Drought-resistant grasses then emerged on the Mongolian Plateau about 7 million years ago during the late Tertiary Period".

Issue 4. Line 63-67: The rationale for this study is fairly clear, but I would suggest the authors moderate their language, especially with the use of "direct" evidence. Phylogenies and biogeographic models are built on a series of assumptions that enable us to infer the history of grassland expansion in relation to climate and landscape events; although these are often reasonable assumptions, this does not equate direct evidence. Direct evidence would comprise temporal and geographic series of grass fossils/biomarkers/etc., which is beyond the scope of this paper.

Response: We agree with the reviewer and appreciate the valuable comment as follows "inference based on phylogenies and biogeographic models is not equally with direct evidence". We will no longer use "direct", and further ask for a native English speaker to check the revised manuscript.

Issue 5. Lines 69-80: I am not familiar with the composition of grassland ecosystems in China – what do the authors mean, for example, by "constructive species in the typical grass- lands"? And when the authors describe Stipa species as dominant, what does this mean (abundance, diversity)? For readers unfamiliar with Chinese grasslands or Stipa more generally, what is a "typical" grassland (line 74) and what is the broader context for Stipa evolution/biogeography? What is the basic for the statement that "the evolutionary processes of Chinese grasslands are closely related to the evolution of Stipa species"? Can the authors describe more about regional grassland communities? Otherwise, Stipa species are just a case study of one group of grasses, rather than a proxy for all grasslands.

Response: We apologize for the confusion. We will add more information about Chinese grasslands and Stipa species as follows: (1) Constructive species is also called edificato, or edificator species. It is the most dominant species in a community, and also plays a significant control role in community structure and function. Stipa species, as the constructive of grassland community, play very important roles in the total Chinese grasslands. However, due to the difference of precipitation and temperature, stipe species show obvious geographical replacement. Thus, the history of origin, divergence and expansion routes of Stipa species are an excellent proxy to reveal the evolution of Chinese grasslands. (2) Chinese grasslands contain as series of grassland types, and typical grassland is one of them. The typical grassland distributes with precipitation of 250 to 400 mm. Meanwhile, typical grassland reflects most obvious grassland feature with dominant drought-resistant perennial herbs.

Issue 6. Lines 86-88: Please include references for BEAST and RASP.

Response: Agreed. We will add references for BEAST and RASP.

Issue 7. Line 139: Can you clarify how FigTree provided the divergence times for Stipa species. I was only aware that this program helped visualize trees.

Response: We apologize for the confusion. We will add more information to clarify how

[Figure]

FigTree provide the divergence times for Stipa species.

Issue 8. Lines 147-148: Why did you assign a maximum number of distribution areas as 2? Please explain your rationale – is that the maximum number of areas any given extant species occurs in?

Response: Due to the difference in precipitation and temperature, the distribution of Stipa species show obvious geographical replacement. Considering that some Stipa species may be emergence on two consecutive areas, thus we assigned a maximum number of distribution areas as 2. The maximum number of areas any given extant Stipa species has been occurred in.

Issue 9. Line 161: What is meant by "the divergence time of Stipa species" in this sentence – I know what you mean when looking at the figure, but in the text it is unclear that this refers to the basal-most split in the studied clade. Since Stipa is a large, widely distributed group, what does this mean in the context of the whole group?

Response: We also apologize for the confusion and agree with the point. Considering Stipa is a large, widely distributed group in the global and the number of Stipa species is only a part, we will revise this sentences as the suggestion.

Issue 10. Line 162: I'm curious as to why the authors did not test for radiations/diversification rate shifts directly. If the primary evidence for an "explosive rapid radiation" is from visual inspection of the tree, I am somewhat unconvinced. It would be better to do a formal test for rate variation, such as using BAMM (Rabosky 2014 in PLOS One). If this kind of test is beyond the scope of the paper, I suggest reframing as an inference/hypothesis that could explicitly tested at a later date.

Response: We thank the review for the constructive suggestion. We will add the BAMM analysis to verify explosive radiation according to the literature(Rabosky 2014).

Issue 11. Line 168: I'm curious what the authors mean by isolated divergences – are they referring to a vicariance model? Can the authors clarify, and justify how they
determined an event to be an isolated divergence event?

Response: We apologize for the confusion. The isolated divergences referred to a vicariance model. In our study, we choose the GTR+G. model to infer the vicariance event based on software jModeltest. We will add more information in the methods part.

Issue 12. Line 171: It is unclear what the authors are referring to – stronger interaction than what? What kind of interaction? This brings up another point – for readers unfamiliar with the landscape of the 7 regions, the topographic setting may be unknown and thus hard to know what the strength of the elevational gradient from the "top and bottom" of the mountains means. Please provide a little more information about the regions (perhaps earlier in the Introduction?).

Response: We also apologize for the confusion. The "interaction" means gene flow. Stronger should be intense. The Qilian Mountains which are important distribution area of Stipa, include many mountain ranges with very different altitudes from about 3000 m to 5000 m. The revised sentences as follows "There was intense gene flow between the top and bottom of the Qilian Mountains including many mountain ranges with very different altitudes from about 3000 m to 5000 m". We will also provide more information about the religion in the introduction part.

Issue 13. Line 182: Can the authors please explain how Figure 4 was generated? Does RASP provide frequency estimates of event types, and from there, authors generated what looks like a kernel density plot? In Figure 4, there is a "Standard" and "Extinction" line; however, these are not mentioned in the text. Can the authors please elaborate? Furthermore, the colors in the figure do not match the legend provided, so it is difficult to align what is in the text with the figure. Finally, the three geologic events illustrated in this figure seem to occur instantaneously, when in reality these events likely took >1myr. Instead of drawing a single line, I suggest providing the age range of geologic events (similar to how it is presented in the text as a range and not a single date).

Response: We are grateful the reviewer for the valuable suggestion and also apologize

for the confusion. The Figure 4 is generated by BEAST software. We will add the details in the revised manuscript. Based on careful review, we also found the colors in the figure do not match the legend provided. We will modify the colors of four lines for a clear view. Meanwhile, we also will provide the age range of geologic events as the text in the figure as the suggestion.

Issue 14. Line 185-187: Can the author please clarify what is meant by "isolation events" – is this vicariance, founder event? Seems like the authors mean vicariance, but it would be helpful to be as clear as possible, and for the terminology in the manuscript to match that of the figures. Furthermore, I'm not sure that I see how the peak frequency (I'm assuming this is frequency, although the y-axis is not labeled) in dispersal and vicariance are "basically matched" in time - can the authors conduct a statistical test to verify this?

Response: We apologize for the confusion. The isolation events means vicariance event in our study. We will modify it in the entire manuscript including figures. Because of the mismatch of colors of Figure 4 and the legend as the comment 13, it is hard to see the peak frequency of the vicariance events. To our knowledge, there is no reasonable statistical test to verify the "basically matched", it is inferred by the evolution process curve (Figure 4).

Issue 15. Line 195-198: This is an incomplete sentence, please rewrite.

Response: We apologize for the mistake. We will revise this sentence as follows "During the second and the third uplift periods of the Qinghai-Tibet Plateau, many species groups originated and differentiated".

Issue 16. Lines 212: Please include a reference for the changing climate conditions.

Response: Agreed. We will add a reference for the change climate conditions as follows (Luo et al. 2016).

Issue 17. Lines 214: I don't see this (short internodes) very clearly; again, this could

be formally tested. Until it is tested, I am unconvinced that this is a clear indication of an explosive, rapid radiation, especially given the large error bars on divergence age estimates. I recommend that the authors moderate their statement – perhaps point to a suggestion of a radiation, but that this remains untested.

Response: We also apologize for the confusion. As the reply to comment 10, we will add the BAMM analysis to verify explosive radiation according to the literature(Rabosky 2014).

Issue 18. Lines 221: There are a few qualitative statements here that I think can be removed: "Due to the crumpling effect of the uplift of the Qinghai-Tibet Plateau, the Tian Shan, Quilian, Altyn Tagh, and Kunlun Mountains all had large-scale elevation of fault-blocks, and many areas that were already elevated became medium-height mountains with around 4000m height."

Response: Agreed. We will remove the qualitative statements of the sentences in the revised manuscript.

Issue 19. Line 225: What do the authors mean here? I suggest rephrasing to say that geographic isolation was likely. "Obvious" here in and elsewhere in this paragraph is kind-of a loaded term, and I suggest avoiding it.

Response: Agreed. We will revise "obvious" to "likely" as the suggestion.

Issue 20. Line 230/235: Is there evidence, other than the assumed absence of physical barrier, for ecological and/or sexual speciation? Ancestral state reconstruction of geographic areas helps inform geography of speciation, but not necessarily mode of speciation.

Response: We agree with the reviewer that ancestral state reconstruction of geographic areas helps inform geography of speciation, but not necessarily mode of speciation In addition to physical barrier, species habitat and phonological differences are possible evident. We will add more information about this discussion in the revised

manuscript.

Issue 21. Line 237-239: This statement is too strong, better to be cautious and use language, such as we infer x or evidence supports y: : : I don't think we can know this definitively, even with a formal test for rate shifts.

Response: Agreed. We will revise it as the suggestion.

Issue 22. Line 240-244: This is an interesting idea. However, I am curious, if Stipa species have a high A+T content and A+T bonds are more prone to mutations, does this imply that the average evolutionary rate of herbaceous plants may be an under-estimate of rates for the Stipa group? If so, how might that affect your findings? In general, a more developed discussion of the assumptions that went into the BEAST and RASP analyses would be good, as well as an explanation for the wide error bars on the reconstructed phylogeny.

Response: We thank the reviewer for the valuable suggestion. On one hand, we will adopt a rate of chloroplast evolution of Stipeae instead of average evolutionary rate of herbaceous plants to calculate the divergence times of Stipa species from this literature (Romaschenko et al. 2014). On the other hand, we will add some discussion about wide error bars on the reconstructed phylogeny of Stipa species.

Issue 23. Line 246-247: It is a little unclear what the division between discussion sections 4.1 and 4.2 is, since geologic and climate history is brought up in 4.1 in relation the divergence dates and geographic expansions. This is up to the authors, but perhaps it would better serve the reader to include some of the background information about the landscape history of the Tibetan Plateau and regions of the study in the Introduction of the paper. Then, the authors could more freely discuss this history throughout the manuscript's discussion.

Response: We appreciate the suggestion. We will revise it as the suggestion.

Issue 24. Line 248: I suggest removing the line "During the developmental process

of the whole geological history" since it is a little unclear what this refers to (e.g., the scope of geological history is far greater in space and time than what is explored in this study).

Response: Agreed. We will remove the line "During the developmental process of the whole geological history".

Issue 25. Line 266-268: Can the authors elaborate on how Oligocene faunal turnover (to a rodent-lagomorph dominated fauna) supports their inferred Miocene Stipa expansion? It is still somewhat unclear, to me, how extensive Chinese grasslands were prior to divergence and expansion of the Stipa and/or whether Stipa are a major player in history of grassland expansion. Or, if they are an interesting group to study because of their dominance (?) today and history in relation to more recent (e.g., Miocene) geologic/ climate events. I think this remains unclear throughout the manuscript – for example, in Lines 291-292, the authors surmise that the uplift of the Qinghai-Tibet Plateau and climate changes promoted the origin of grasslands, which appears to contradict an earlier origin inferred from faunal turnover and mentioned in Line 266. This confusion might be cleared up by clarifying early on the current state of knowledge (based on fossil evidence, other non-Stipa groups, etc.) and how Stipa specifically contributes to the grassland story in China – e.g., does it signal grassland expansion?

Response: We appreciate the constructive suggestion and agree with the point. Considering the grassland emerged in Inner Mongolia Plateau at 33 MaBP (Meng and McKenna 1998) and Stipa species originated at 28 MaBP from our finding. It may be that Stipa species was not the most primitive species of the grassland. It may have evolved after the respective grasslands and subsequently invaded and became dominant. Thus, we will find more fossil evidence to prove the status of Stipa species in the history of grassland evolution. Otherwise, we will restate the goals of the study to focus on the evolutionary history of Stipa without the assumption that the history of Stipa is a good proxy for the evolution of the grasslands.

Issue 26. Line 284: Suggest replacing "outbreak" with expansion, shift in ecological dominance,etc:

Response: Agreed. We will replace "outbreak" with expansion as the suggestion.

Issue 27. In general, the text in the accompanying figures is small and difficult to read. Is it possible to enlarge the figures and figure text?

Response: We apologize for the low quality figures. We will re-draw all the figures for enlarging the figures and figure text.

Issue 28. Figure 2: There are very wide error bars on divergence times; this should be mentioned in the results and should be discussed in detail in the results and/or discussion. What contributes to wide error on divergence age estimates and how does this influence your interpretations of evolutionary processes? Stipa should also be capitalized in the Latin names.

Response: Thanks the reviewer for the valuable comment. As the reply to comment 22, we will add some discussion about wide error bars on the reconstructed phylogeny of Stipa species. Meanwhile, we will modify stipa with Stipa.

Issue 29. Figure 3: Can you make this figure larger? It is difficult to read as is, especially the ancestral states and landmark nodes on the phylogeny. Furthermore, the colors on the phylogeny seem to correspond with different regions. Can you color code the different regions on the map as well? Provinces appear to contain multiple biogeographic regions, so it is difficult to tell where the region boundaries are. Not necessary, but it might also help get the authors' message across if another panel is included with terrain, so that the readers can know where mountains ranges exist, etc. in relation to the biogeographic regions and inferred dispersal routes.

Response: We are grateful for the constructive suggestion. We will re-draw the figure as the suggestion including enlarging the figure, coloring the map, deleting the province lines and adding the terrain.

Issue 30. Figure 4: Please add y-axis and x-axis labels to this figure, and more detail as to how this figure was constructed. Are we looking at output from the RASP analysis? In addition, please change the colors of the curves to match those of the figure legend (for example, I cannot tell which curve is the extinction and which is the standard). In the figure caption, what does a time-geological time curve mean? Do the authors simply mean event curves over geologic time from 30 Ma to present? And, is it necessary to use the terminology "time abscissas and range ordinates"? I think this is confusing, when, I believe, the authors are just describing x and y coordinates.

Response: We apologized for the confusion We will re-draw the figure as the suggestion including adding x-axis and y-axis labels, changing the color of lines. A time-geological time curve means evolution process curve of Stipa species and corresponding geological events. As the reviewer elaborated, it is not necessary to use the terminology "time abscissas and range ordinates", it is enough to describe x and y coordinates.

Issue 31. Frequent typos: e.g., missing spaces between word and reference, missing punctuation; inconsistent pluralization of grasslands, area, etc.; "stipa" is lower case in the figures; comparative adjectives used without a comparison noun (e.g., lines 182-183 – "larger" should be "large" or the authors should state what the expansion is larger than); "MaBP" can just be "Ma"

Response: We apologize for all the mistakes and appreciate the review for careful checking. We will read through the manuscript to correct all the corresponding mistakes.

References

Luo D, Yue JP, Sun WG et al. 2016. Evolutionary history of the subnival flora of the Himalaya-Hengduan Mountains: first insights from comparative phylogeography of four perennial herbs. J. Biogeogr., 43(1):31-43.

[Figure]

Meng J, McKenna MC. 1998. Faunal turnovers of Palaeogene mammals from the Mongolian plateau. Nature, 394(6691):364-367.

Rabosky DL. 2014. Automatic Detection of Key Innovations, Rate Shifts, and Diversity-Dependence on Phylogenetic Trees. PLoS One, 9(2)

Romaschenko K, Garciajacas N, Peterson PM et al. 2014. Miocene-Pliocene speciation, introgression, and migration of Patis and Ptilagrostis (Poaceae: Stipeae). Molecular Phylogenetics & Evolution, 70(1):244-259.

Please also note the supplement to this comment:
https://www.biogeosciences-discuss.net/bg-2018-140/bg-2018-140-AC3-supplement.pdf
* * *